# Towards effective clinical decision support systems: A systematic review

**Francini Hak** *, **Tiago Guimarães**, **Manuel Santos**

Algoritmi Research Center, University of Minho, Braga, Portugal

* francini.hak@algoritmi.uminho.pt

## Abstract

### Background

Clinical Decision Support Systems (CDSS) are used to assist the decision-making process in the healthcare field. Developing an effective CDSS is an arduous task that can take advantage from prior assessment of the most promising theories, techniques and methods used at the present time.

### Objective

To identify the features of Clinical Decision Support Systems and provide an analysis of their effectiveness. Thus, two research questions were formulated: RQ1—What are the most common trend characteristics in a CDSS? RQ2—What is the maturity level of the CDSS based on the decision-making theory proposed by Simon?

### Methods

AIS e-library, Decision Support Systems journal, Nature, PlosOne and PubMed were selected as information sources to conduct this systematic literature review. Studies from 2000 to 2020 were chosen covering search terms in CDSS, selected according to defined eligibility criteria. The data were extracted and managed in a worksheet, based on the defined criteria. PRISMA statements were used to report the systematic review.

### Results

The outcomes showed that rule-based module was the most used approach regarding knowledge management and representation. The most common technological feature adopted by the CDSS were the recommendations and suggestions. 19,23% of studies adopt the type of system as a web-based application, and 51,92% are standalone CDSS. Temporal evolution was also possible to visualize. This study contributed to the develop-ment of a Maturity Staging Model, where it was possible to verify that most CDSS do not exceed level 2 of maturity.

**Data Availability Statement:** All relevant data are within the manuscript and its Supporting information files.

**Funding:** This study was supported by FCT – Fundação para a Ciência e Tecnologia, within the

scope of the project DSAIPA/DS/0084/2018, FH work was supported by the grant 2021.06230.BD. The funders had no role in study design, data collection and analysis, decision to publish, or preparation of the manuscript.

**Competing interests:** The authors have declared that no competing interests exist.

## Conclusion

The trend characteristics addressed in the revised CDSS were identified, compared to the four predefined groups. A maturity stage model was developed based on Simon's decision-making theory, allowing to assess the level of maturity of the most common features of the CDSS. With the application of the model, it was noticed that the phases of choice and implementation are underrepresented. This constitutes the main gap in the development of an effective CDSS.

## Introduction

In the day-to-day routine of healthcare units, the domain professionals interact with hundreds of patients. Naturally, it is of great importance that all this interaction is transformed into records, whether clinical or administrative, which in turn are transformed into data, information, and, hopefully, knowledge. Despite technological developments, these records are still kept in physical format, which creates a delay in the work performed when compared to digital versions [1].

Friedman [2] boasted a theorem in which it shows that an individual or group working with the contribution of a technological resource of information, has a better performance compared to a job without such assistance. For this functional theorem to be verified, the information resources must be valid and reliable and the user must know how to handle it properly.

The electronic representation of clinical records are embodied by Electronic Health Records (EHR), which aim, in particular, to eliminate the use of paper in the healthcare field. Notwithstanding, the representation of electronic health data tends to be more than just a substitute for paper [1]. HIMSS [3] shows that a healthcare information system must have the ability to generate a complete view of the clinical record at a meeting with a patient, including decision evidence-based support, quality management and results reporting, apart from the support of other activities related to direct or indirect care by means of a shared area. Thereby, it is evident that for a healthcare information system to be effective, it is crucial the existence of an integrated component of support in decision-making processes.

The concept of Decision Support (DS) is equivalent to the activity that assists a given user following a certain purpose. Thus, a Decision Support System (DSS) turns this activity into a system-based format in an efficient and reliable way, composing models and techniques through a knowledge representation infrastructure [4]. In this sense, a DSS portrays a system designed to support a professional in obtaining knowledge and making decisions in the specific area, therefore, diminishing uncertainties during the decision-making process [5].

In the healthcare field, the decision-making support activity is designed as Clinical Decision Support (CDS). The representation of this activity in a computerized-based format is translated into a Clinical Decision Support System (CDSS), where it provides all the information and desired knowledge that facilitate the daily tasks of healthcare providers and guarantee an improvement in the quality of services.

As a clinical knowledge-based system, the treatment of information and the process of knowledge extraction are considerable aspects for a Clinical Decision Support System to achieve its objectives. Some features are mandatory in a CDSS, but not all are known and some may be missed and contribute to system failures [6].

According to Simon, the decision-making process is the heart of any organization and it influences all processes integrated into it [7]. The decision-making process can be outlined using various models and theorems. Within the scope of this study, the Bounded Rationality theory was adopted as a reference to Herbert Simon's work [8]. It is stated that people act in a rational way according to the knowledge and perception that they get. In an initial phase, this model approached three stages: Intelligence, Design and Choice. Later on, a new stage was added by Sprague and Carlson [9], resulting in the implementation phase. Thus, the four phases resulted by Simon's work [8], redound to model decision making process of an Intelligent Decision Support System.

Previous systematic review studies (SRs) have addressed the use of a Clinical Decision Support System to assess its use and effectiveness [10–12], but applied to a specific workflow. Instead, this systematic review aims to analyse a set of articles that address a CDSS in order to identify the trends of the characteristics for its conception. In addition, we also identify other aspects such as the purpose of care and the recipient of the intervention. Furthermore, in order to enrich the outcomes obtained from the review, we trace a general trend of a CDSS, relating it with the four phases proposed by Simon.

This article is structured in four sections. First, an introduction to the topic of study is presented. Secondly, the landscape of the methods used to make the Systematic Review. The third section presents the results of the Systematic Review. The fourth section is dedicated to discussing the results obtained. At last, in section five, conclusions are drawn while leaving open doors for future work.

## Decision-making theory

Decision-making is one of the main processes dealt within an organization [13]. This process can be approached using different models and theories. In the present study, the bounded rationality theory proposed by Simon will be focused.

Herbert Simon [8] was an economist who carried out research involving several areas, such as psychology, computing and management. One of Simon's great contributions to scientific research was the theory of bounded rationality, where Simon redefines human rationality arguing that people act rationally according to the knowledge and perception they obtain. To formalize his theory, Simon established four phases that define the decision-making process, as shown in Fig 1.

In Intelligence phase, problems are identified and the purpose of action involving the decision is determined. In the Design phase, possible solutions are designed and alternatives are created to solve the problem. Consequently, in the Choice phase, the alternative that most satisfies the purpose of action will be chosen. Years later, Turban [14] was responsible for reformulating Simon's theory by adding a fourth phase in order to assess the implementation process. Thus, the Implementation phase results from the joining of the three previous phases

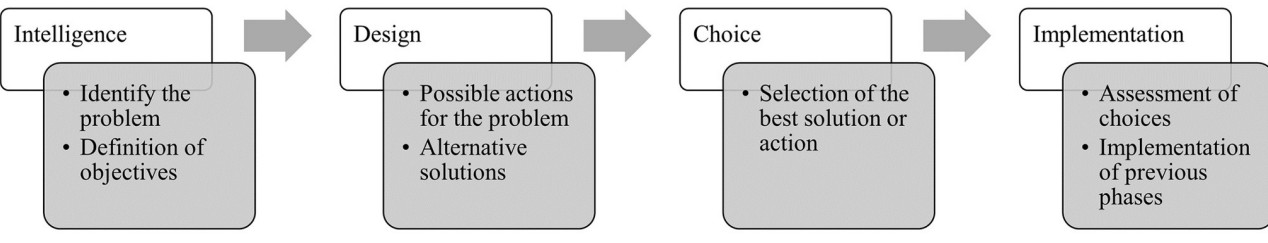

**Fig 1. Iterative phases of the decision-making process.**

put into practice. It should be noted that the decision-making is a sequential and iterative process, which requires the completion of all phases to reach the final phase and an ongoing review of the same.

- Intelligence—Process of formalization of the problem and definition of decision conditions. The decisions to be made with the help of the proposed system and the benefits it will bring are still unclear and not understood. In this phase, reality is examined in order to model the existing information for decomposing the problem and determining its properties. The intelligence phase ends with a problem statement.

- Design—At this stage it is necessary to have the problem defined to search for alternatives or available options, where possible courses of action are analysed and validated. Problem-situation models are built to explore alternative solutions. For this, it is necessary to have well-defined and declared selection criteria as a result for the evaluation of potential solutions that will be identified (includes techniques, technologies, format, integration, functionalities, etc). Several possible outcomes can be considered for each alternative, each with a certain probability of occurrence. In cases where decisions are made with uncertainty, there are multiple outcomes for each alternative.

- Choice—In the choice phase, all alternatives are researched, evaluated, and one is defined as the recommended solution. In pursuit of goals, a course of action that is good enough is selected. Computer models or critical success factors can be used as techniques to evaluate the recommended solution. Normative models can use either the analytical model or an exhaustive and complete enumeration model. The solution is tested and once the proposed solution appears viable, a decision can be made.

- Implementation—This phase is the validation of all previous phases. If the adopted solution seems good enough, it can be implemented. Successful implementation results in solving the original problem. In case of failure, it must be returned to the previous phases.

## Methods

The systematic review was conducted based on Preferred Reporting Items for Systematic Reviews and Meta-Analysis (PRISMA) statements [15], followed by a checklist and a flow diagram.

### Information sources

Five online data sources related to research in health information systems were selected. First, a search was performed on Scimago [16] comprising two areas of study: information systems and multidisciplinary. From this, journals with Q1 quartile in the last 5 years and with an impact factor greater than 150 were analysed. Journals that had the potential to have articles more related to the purpose of this study and that were already known by the authors were selected: Decision Support Systems, Nature Research Journal and Plos One. To complement the study, two public repositories were chosen, such as: the PubMed library (medline) and the Association of Information Systems (AIS) e-library. The Table 1 shows information about the selected data source types.

### Eligibility criteria and search strategy

To select the studies for the development of the systematic review, eligibility criteria were defined: (i) published studies in article format; (ii) open access or free full text; (iii) written in

**Table 1. Information of selected data sources.** Source: Scimago Journal and Country Rank via www.scimagojr.com, accessed on May 13, 2022.

| Data Source | Publication Type | Subject Area | Quartile | H-Index |
|---|---|---|---|---|
| AIS e-library | Repository | Information Systems | - | - |
| Decision Support Systems | Journal | Information Systems | Q1 | 161 |
| Nature | Journal | Multidisciplinary | Q1 | 1276 |
| Plos One | Journal | Multidisciplinary | Q1 | 367 |
| PubMed | Repository | Biomedical | - | - |

English; (iii) articles addressing a decision support systems in healthcare; (iv) computerized or electronic decision support systems. To meet these criteria, search filters were applied. The selected articles must address practical cases of a specific clinical decision support system, therefore, articles of the literature review type or that did not address a particular CDS system were excluded.

The search strategy was performed considering an ordering from the most recent publications to the oldest ones, restricting to a range from 2000/01/01 to 2020/12/31. To identify studies of the desired scope, the query terms were: ((everything:"Clinical Decision Support System") OR everything:"Clinical Decision Support") OR everything:"Decision Support System in Healthcare". The title, abstract and keywords were the primary strategy for analyzing each scientific publication. When these parameters were not sufficient, a complete reading of each article was made, in order to guarantee the eligibility of inclusion criteria.

### Data extraction and management

The screening process was carried out by two authors (FH and TG) independently, by reading the titles, abstracts and keywords of the publications of the search results. When the criteria were met, the full text was read. In moments of disagreement or doubt, the third reviewer (MS) took a stand and contributed to the discussion.

Through the review of eligible articles, data extraction was performed by two reviewers (FH and TG) and focused on four characteristics of a CDSS, defined as: knowledge management and representation technique; technological resources integrated into the system; the type of system; and system integration. These four categories were considered by the authors as the most relevant for the purpose of this study. In addition, the following information was extracted: clinical setting, study design, recipient of intervention, and purpose of care. Data extraction was performed through the textual analysis of the articles and the search for keywords. For better management and analysis, the extracted data were recorded on an appropriate sheet (see S1 File).

To answer the second research question of this study, it was identified in which phase of the decision-making theory the CDSS of the reviewed articles was found. Thus, based on the literature on decision-making theory [17, 18] and the definitions of each phase, the authors considered a set of criteria for the assignment of each phase, as shown in Table 2. For a system to be classified in a certain phase, it must fulfill at least one of the conditions indicated.

### Quality assessment and data synthesis

All the reviewed studies portray a Clinical Decision Support System (CDSS) designed for a specific purpose. The key points for conducting this systematic review were to identify the trends of the CDSS developed regarding their type of system, integration, representation and formulation of knowledge, and their technological features.

**Table 2. Choice of criterias for the classification of Simon's phases.**

| Phase | Condition 1 | Condition 2 | Condition 3 | Condition 4 |
|---|---|---|---|---|
| **Intelligence** | Have access to the history of past decisions. | Ability to predict what may happen to the clinical situation (e.g. with machine learning and data mining models). | Having structured and available information capable of accessing specific clinical information for the characterization of the situation. | Allow access to data to assess the clinical situation of patients. |
| **Design** | Definition of clinical context variables. | Ability to develop and present alternative solutions to a clinical situation. | Use of modeling techniques/procedures. | Use of models or set of models to learn to decide. |
| **Choice** | Be able to simulate and evaluate each of the available alternatives. | Choice of best alternative as recommended solution based on applied criteria or models. | Application of computational models to ensure the operation of the chosen solution. | - |
| **Implementation** | Application of the chosen option in a real environment. | Monitoring and evaluation. | Data collection from monitoring and evaluation. | Be able to record past decisions. |

Note: The conditions presented are of disjunction, that is, at least one of them must be met for the system to be classified in a certain phase.

The reporting quality was made following the PRISMA checklist. Publications considered eligible for the analysis were assessed for the methodological quality, meeting the inclusion criteria, risk of bias, assumptions and simplifications, and clarified evidence-based results. The reviewers did not apply any methodological assessment tool.

After defining the five data sources, the reviewers selected a set of publications within the scope of Clinical Decision Support Systems, through the search strategy and the inclusion criteria previously defined. The outcomes presented through tables and narrative summary characterize the different Clinical Decision Support Systems presented in the selected studies, in order to trace a global trend. The data from the matching results were pooled out and evaluated, based on the total value of studies and the probability of occurrence. The reviewers noted the presence of clinical and statistical heterogeneity among the studies.

## Results

### Study selection

The study selection was carried out following the PRISMA statement [15], using a checklist (see S2 File) and a flow diagram represented in Fig 2 (S3 File). 768 records were identified by searching five databases considered most influential for the purpose. After removing duplicate records, 690 records were screened to read the title, abstract and keywords. Lastly, the articles that did not meet the selection and search criteria were excluded, comprising: non-article format; unpublished articles; not in english; out of period; not open access. The selected articles were read in full to assess their eligibility. Finally, 52 articles met the research questions and were elected eligible and appropriate to carry out the systematic review, embracing a specific Clinical Decision Support System and not literature review articles.

### Study characteristics

The studies that met the inclusion criteria were characterized according to a set of attributes. The period covered between the studies was from 2000 to 2020, with 50% from 2020, giving priority to the most recent ones.

The studies were searched in the five previously selected data sources, with eighteen studies from PubMed (34,62%), eleven studies from Decision Support Systems journal (21,15%), ten studies from PlosOne (19,23%), and five studies from Nature (9,62%). We note a diversity in

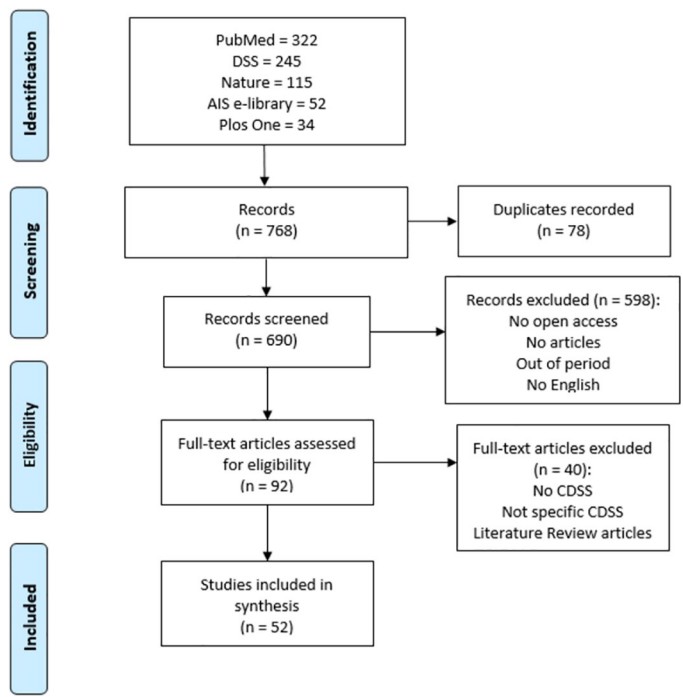

**Fig 2. Flow diagram of study selection for systematic review.**

the countries that developed the studies, covering four different continents. However, the majority stand out to the United States of America, representing nineteen studies (34,62%).

Studies realized in any healthcare setting were the most frequent ones (28,85%), followed by hospital setting (21,15%) and hospital academic centers (21,15%). The most common recipients of the intervention are, directly, general practitioners (32,69%). Studies reveal that CDSS also has an action on patients, but mostly, indirectly.

In general, the study design of the reviews articles, demonstrates the evaluation of the effectiveness of the CDSS regarding its cost, implementation and usability. In addition, the review showed that the CDSS addressed have an associated clinical intervention, with the most common purpose of care being a specific workflow (32,69%), followed by a specific patient disease (15,09%).

## Outcomes

The information extracted from the studies that were considered relevant for the desired purpose generated a set of outcomes, based on the characterization of a Clinical Decision Support System (CDSS).

**Knowledge management**. The representation and formulation of the acquired knowledge is one of the most important steps in the development of a CDSS. In order to identify the most used techniques for knowledge management, we have identified the approaches used in studies of the systematic review, as shown in Table 3. Clinical Practice Guidelines (38,46%), rule-based module (40,39%) and algorithmic logic (38,46%), were the most used approaches to knowledge management in the designing of a CDSS. These techniques were used individually or in combination with others. The formulation of a knowledge base was also identified in thirteen studies (25%), in combination with other techniques. Inference engines were

**Table 3. Outcomes of knowledge representation and management.**

| Study | Outcome | Total (n) | Value (%) |
|---|---|---|---|
| [21–41] | Rule-based module | 21 | 40,39% |
| [21, 22, 30, 40–56] | Clinical Practice Guidelines | 20 | 38,46% |
| [21, 24, 26, 29, 32–34, 37, 39, 42–44, 49, 57–63] | Algorithmic logic | 20 | 38,46% |
| [24, 25, 28, 30, 31, 34, 37, 38, 40, 51, 52, 64, 65] | Knowledge base | 13 | 25% |
| [28, 35, 52, 57, 60, 66–68] | Variable-based | 8 | 15,39% |
| [25, 26, 30, 40, 51–53] | Inference engine | 7 | 13,46% |
| [27, 31, 68–70] | Standardized Clinical Terminologies | 5 | 9,62% |
| [35, 38, 57] | If/then statements | 3 | 5,77% |
| [29, 33, 39] | Data mining techniques | 3 | 5,77% |
| [48, 57] | Bayesian network | 2 | 3,85% |
| [67, 68] | Neural networks | 2 | 3,85% |

identified in seven studies (13,46%), which were also considered in some studies, as reasoning engines. Three studies (5,77%) provided if/then statements, and eight studies (15,39%) applied methods based on variables. Terminology standards and clinical classification systems were present in five studies (9,62%), applying, in particular, the ICD-10 system and the HL7 communication protocol. Bayesian and neural networks were used in two different articles (3,85% each). Finally, three studies (5,77%) used data mining techniques to reproduce the desired knowledge. Two studies [19, 20] were excluded from pooling, as the method used in the knowledge representation process was not identified.

**Technological features**. Technological interventions represent the features that contribute to the system achieving its purpose. The most common technological feature approached in the CDSS is recommendation and suggestion feature, identified in twenty four studies (46,15%), as shown in Table 4. Information management and monitoring are the second most common feature in the systems, covering eighteen studies (46,15%). The third most desired feature is alerts, notifications and reminders, covering fourteen studies (26,92%). Therefore, it follows the purpose of reducing errors as mentioned in eleven studies (21,15%). Eight articles (15,38%) design the CDSS for assessment purposes. The prediction feature is also used for

**Table 4. Outcomes of technological features.**

| Study | Outcome | Total (n) | Value (%) |
|---|---|---|---|
| [20, 22, 25, 29, 30, 36, 38, 40–42, 46, 47, 50–55, 57, 60, 61, 64, 67, 68] | Recommendation and suggestion | 24 | 46,15% |
| [27, 28, 35, 37, 39, 40, 43, 44, 45, 48, 50, 53, 56, 58, 62, 63, 66, 70] | Information management and monitoring | 18 | 34,62% |
| [23, 30–32, 34, 37, 38, 42, 43, 45, 47, 49, 50, 53] | Alerts, notifications and reminders | 14 | 26,92% |
| [19, 21, 32, 37, 38, 43, 48, 50, 51, 68, 69] | Error reduction | 11 | 21,15% |
| [24, 25, 36, 45, 50, 57, 59, 66] | Assessment | 8 | 15,38% |
| [33, 35, 50, 59, 60, 67, 68, 70] | Prediction | 8 | 15,38% |
| [27 32, 36, 46, 53, 61, 65] | Process automation and prioritization | 7 | 13,46% |
| [31, 34, 48, 65] | Events | 4 | 7,69% |
| [23, 26, 61, 69] | Standardization | 4 | 7,69% |
| [19, 52, 59] | Calculation and scoring | 3 | 5,77% |
| [21, 23] | Cost and time reduction | 2 | 3,84% |

**Table 5. Outcomes of system integration.**

| Study | Outcome | Total (n) | Value (%) |
|---|---|---|---|
| [20, 24, 25, 29, 31, 35, 36, 40, 41, 42, 44, 47, 51–55, 58–65, 69, 70] | Standalone CDSS | 27 | 51,92% |
| [19, 21, 37, 39, 43, 45, 46, 49, 57, 66, 68] | Electronic Health Record (EHR) | 11 | 21,15% |
| [26–28, 30, 33] | Specific Healthcare Information System | 5 | 9,62% |
| [23, 34, 38, 48] | Computerized Provider Order Entry (CPOE) | 4 | 7,69% |
| [22, 32, 56] | Computerized Provider Order Entry (CPOE) and Electronic Health Record (EHR) | 3 | 5,77% |
| [50, 67] | Electronic Medical Record (EMR) | 2 | 3,84% |

eight studies for different purposes (15,38%). The automation and prioritization of processes is considered to be of great importance and is addressed by seven articles (13,46%). The triggering of events is addressed by four studies (7,69%). The standardization of the clinical process is desired by four studies (7,69%). Three articles (5,77%) highlighted the calculation and scoring methods as features of a CDSS. Finally, the cost and time reduction is seen as a main feature in two studies (3,84%).

**System integration**. The results related to the integration of the Clinical Decision Support System (CDSS), showed that 51,92% of the articles (twenty-seven) use a standalone system, as shown in Table 5. The remaining studies show that their CDSS are integrated with another system, as well as using the information from these systems. Thus, eleven studies (21,15%) reveal that they integrate their CDSS with an Electronic Health Record (EHR) system; five studies (9,62%) integrate their system with a specific health information system; four studies (7,69%) integrate with a Computerized Provider Order Entry (CPOE) system; in this sequence, three studies (5,77%) integrate their CDSS with both EHR and CPOE; two studies differentiate their integration with an Electronic Medical Record (EMR) system.

**Type of system**. The typology of a Clinical Decision Support System (CDSS) differs in several technological aspects, considering both the presentation of the user interface and the technique it is based on. The type of systems identified in the review of studies is represented in Table 6. Most studies (19.23%) use a web-based application to present the system interface to the user. Nine studies (17,31%) describe their CDSS as a computerized tool, not specifying the type of software or interface used. Eight studies (15,38%) develop their decision support system in a software application. Regarding the logic used, five studies (9,62%) are based on machine learning techniques, and four studies (7,69%) are artificial intelligence-based. Three studies (5,77%) develop their CDSS in a mobile application and, still, four studies (7,69%) cover a mobile and web application. Two studies (3,85%) describe their CDSS as knowledge-based, and other two studies (3,85%) present the CDSS as data analytics. Other types of systems were considered in unique studies, such as cloud computing (1,92%), data-layer infraestructure (1,92%), image retrieval expert system (1,92%), user interface (1,92%), and web application integrated with data analytic (1,92%).

The outcomes for each group were varied regarding the type of the CDSS, knowledge management, integration of systems and the technological features. Despite this diversity, we were able to notice a trend in results based on their frequent presence in the reviewed studies. As shown in Fig 3, the most common knowledge representation and management techniques were the rule-based module, clinical practice guidelines and algorithmic logic. Regarding to the technological intervention or feature, the three top trends were recommendation and

**Table 6. Outcomes of type of system.**

| Study | Outcome | Total (n) | Value (%) |
|---|---|---|---|
| [22, 25, 30, 46, 47, 49, 50, 51, 65, 66] | Web-based application | 10 | 19,23% |
| [21, 23, 26, 36, 45, 48, 56, 57, 68] | Specific computerized tool | 9 | 17,31% |
| [19, 27, 35, 37, 42, 54, 62, 64] | Software application | 8 | 15,38% |
| [32, 39, 58–60] | Machine learning-based | 5 | 9,62% |
| [20, 33, 52, 70] | Artificial intelligence-based | 4 | 7,69% |
| [34, 44, 53, 69] | Web and mobile application | 4 | 7,69% |
| [29, 31, 61] | Mobile application | 3 | 5,77% |
| [40, 67] | Data Analytics | 2 | 3,85% |
| [41, 55] | Knowledge-based | 2 | 3,85% |
| [38] | User interface | 1 | 1,92% |
| [63] | Cloud computing | 1 | 1,92% |
| [24] | Data-layer infrastructure | 1 | 1,92% |
| [28] | Image retrieval expert system | 1 | 1,92% |
| [43] | Web-based application and data analytics | 1 | 1,92% |

suggestion, information management and monitoring, and alerts, notifications and reminders. Standalone CDSS were the most common one, following integration with Electronic Healthcare Records (EHR) systems and specific healthcare information systems. The most frequent type of system were, respectively, web-based application, specific computerized tool, and software application. Despite the individual results, we also noticed a trend towards a combination of results. The most frequent sequence related to knowledge management was the mix of algorithmic logic with clinical practice guidelines. For technological features, the most common combination was the joining of recommendation and suggestion with alerts, notifications and reminders.

## Temporal evolution

Fig 4 demonstrates the temporal evolution of knowledge management and representation techniques used in studies. It is possible to verify that rule-based module and the algorithmic logic have been present since 2000 until today, mostly. As for clinical practice guidelines, on the other hand, proved to be a current issue due to their presence from 2013 to 2020. In general, the characteristics remain in existence over time, with a greater occurrence in 2020 due to the number of articles analysed that year. In contrast, the characteristics of the Bayesian network, if / then statements and neural networks were considered more recent, emerging from 2014.

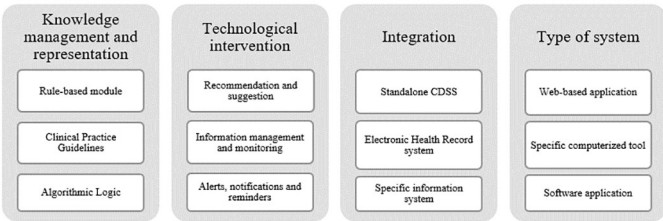

**Fig 3. Characterization of general trends obtained through the review.**

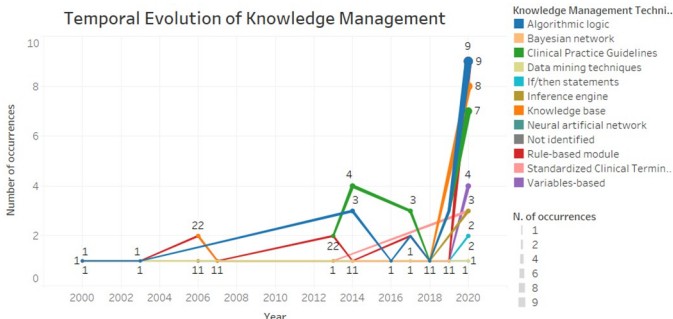

**Fig 4. Temporal evolution of knowledge management.**

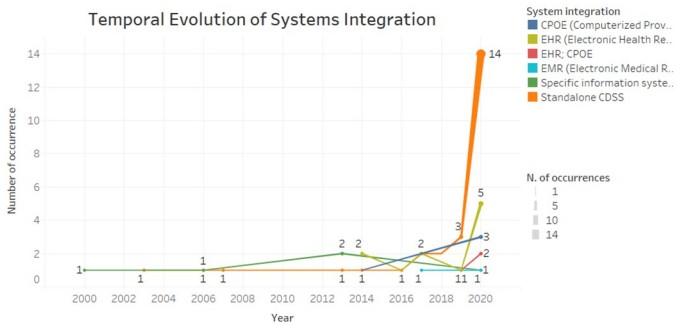

**Fig 5. Temporal evolution of system integration.**

According to Fig 5, we can see that the integration of standalone CDSS has been increasing in recent years. The integration with EHR systems, oscillated between 2014 to 2020, standing out also in the last year. Specific health information systems have been integrated into the CDSS from 2000 to 2020, on a regular basis.

According to Fig 6, the specific computerized tools have been present since the beginning of the time interval and were more present in the year 2020. The software application, machine learning-based, was already present. The mobile application remained constant. Web-based application was more prevalent in 2013.

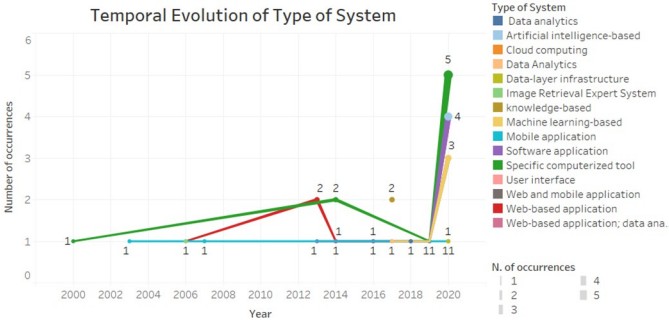

**Fig 6. Temporal evolution of type of system.**

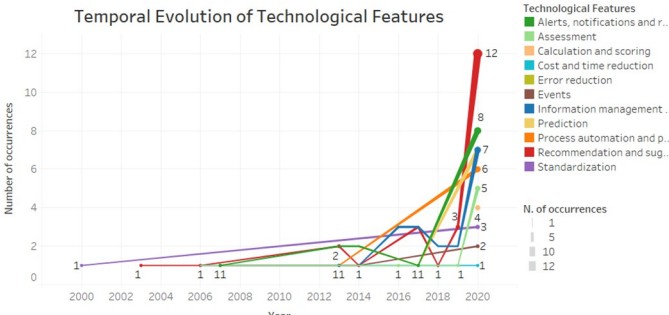

**Fig 7. Temporal evolution of CDSS features.**

As showed in Fig 7 the alerts, notifications and reminders and the information management and monitoring feature were more present in 2020, but also present in some previous year. The recommendations and suggestions feature was present from 2003 to 2020. Standardization was present in 2000 and only returned in 2020. The calculation and scoring, cost and time reduction and prediction features were considered the most recent ones.

## Maturity staging model

For a CDSS to be considered effective, it has to reach the prestigious level of maturity. Thus, it is important to recognize the degree of maturity associated with the characterization of the CDSSs. In order to complement the study, a cross-checking was made with the trends characteristics facing the four phases of Simon's decision-making theory [8]. Thus, four stages were classified aiming to represent the level of maturity of a system, based on Simon's phases:

- Stage 4: Implementation + Choice + Design + Intelligence

- Stage 3: Choice + Design + Intelligence

- Stage 2: Design + Intelligence

- Stage 1: Intelligence

Table 7 represents the crossing of the Simon's phases with the three most common characteristics of each group, in order to assess their stage of maturity. The columns referring to the maturity stages, correspond to the number of articles of the respective characteristic, given their presence in each Simon phase. It should be noted that the values are cumulative, that is, for a stage to be reached, it must contain the previous stage. All studies corresponding to each characteristic are present in the initial phase of Intelligence. Maturity is calculated using the weighted average (WAVG), which should vary between 1 and 4, corresponding to each stage.

## Discussion

### Main findings

This article aimed to develop a systematic review within the scope of Clinical Decision Support Systems (CDSS). The first research question (RQ1) was to identify the tendency of adopted approaches in CDSS development addressed in the reviewed studies, as shown in Fig 3. To classify the revised studies, information about four predefined groups were extracted: knowledge management, technological intervention, system integration and type of system. Although there are other characteristics associated with a CDSS, the authors chose to extract

**Table 7. Maturity Staging Model applied on the reviewed studies.**

| Characteristic | Stage 1 | Stage 2 | Stage 3 | Stage 4 | WAVG |
|---|---|---|---|---|---|
| **Knowledge Representation** | | | | | |
| Rule-based module | 21 | 16 | 4 | 1 | 1.64 |
| Clinical Practice Guidelines | 20 | 12 | 5 | 2 | 1.71 |
| Algorithmic Logic | 20 | 13 | 7 | 2 | 1.79 |
| **Features** | | | | | |
| Recommendation and suggestion | 24 | 13 | 5 | 0 | 1.55 |
| Information management and monitoring | 18 | 9 | 7 | 1 | 1.74 |
| Alerts, notifications and reminders | 14 | 9 | 3 | 0 | 1.58 |
| **System Integration** | | | | | |
| Standalone CDSS | 27 | 16 | 6 | 1 | 1.62 |
| Electronic Health Records | 11 | 6 | 3 | 1 | 1.71 |
| Specific information system | 5 | 4 | 1 | 0 | 1.6 |
| **Type of System** | | | | | |
| Web-based application | 10 | 8 | 2 | 0 | 1.6 |
| Specific computerized tool | 9 | 4 | 2 | 1 | 1.67 |
| Software application | 8 | 6 | 1 | 0 | 1.53 |

specific information that go beyond the main objective of a CDSS, which is to assist the decision maker in the decision-making process. The results showed that the most used techniques for the knowledge representation in the systems are the creation of modules based on rules, clinical practice guidelines and logic algorithms. The technological features most present in CDSS are recommendation and suggestion, monitoring and information management, and alerts and reminders. Usually, CDSS are used standalone, following the integration with Electronic Health Systems or a specific information system. It was also identified that the most common types of systems are web applications, specific computer tools, and software applications.

In order to answer the second research question (RQ2), a preliminary classification during the review was carried out identifying which phase of the decision-making process model the CDSS was in. According to Simon's bounded rationality theory [8], it was considered: Intelligence phase as responsible for analysis, exploration and description of the problem to be faced; the Design phase as the development and analysis of possible solutions to the problem at hand; in the Choice phase, the most appropriate solution is chosen to solve the problem; the Implementation seeks to apply the solution to the problem in question.

Regarding the maturity staging model proposed in Table 7, when the stage 4 is achieved, the system has reached its prestigious level of maturity. However, this is not verified in the analysis. The weighted average (WAVG) of the characteristics showed very similar values, meaning that the CDSS of the reviewed studies predominates in Simon's Intelligence and Design phases, equivalent to stage 2 of maturity. The recommendation and suggestion (1.55) and software application (1.53) characteristics have the lowest values nearly reaching the previous phase of Intelligence. The algorithmic logic characteristic stands out the most and shows that it is closer to achieving the upper stage of maturity (stage 3), meeting the Choice phase.

There are other systematic reviews that study the role of CDSS. However, to the best of our knowledge, existing studies analyse CDSS applied to a specific clinical context and do not evaluate its effectiveness as a whole (as [11, 12, 71]). The present systematic review selected a set of studies that approach the development of a CDSS characterizing them in order to trace a global trend, as well as assess their level of maturity. In this sense, it was possible to analyse the CDSS

in its completeness, covering different approaches, techniques and purposes of use. The main contribution of this work was the proposed maturity staging model, that allowed to identify a gap between the state-of-art and the desirable stage of maturity in order to provide an effective CDSS. The results showed that the revised CDSS do not go beyond stage 2, meaning that CDSS are not succeeding in the healthcare arena due to the lack of maturity, i.e., as CDSS are not capable of supporting the choice of actions in clinical settings and are also not involved in the implementation of these actions.

This study allowed to raise a concern in the development of CDSS and to raise awareness that limited systems are being created and that may be far from being optimized. The projection of this study allowed us to portray a reality of many decision support systems in the health area, demonstrating the opposite of what it should be. When a system reaches the model implementation phase, it should be closer to reaching its optimization and not going back to the previous phases. This immaturity may be due to a lack of understanding of the real problem, difficulty in choosing the ideal solution, and failures in usability tests. An in-depth analysis must be done to discover the main constraints that prevent the inclusion of the decision-making phases in the CDSS.

## Limitations and future research

The maturity staging model, combined with the phases of the decision-making process, serves to assess the effectiveness of the decision. When an implemented decision does not produce the desired results, there are likely to be several causes, such as incorrect problem definition, poor evaluation of alternatives, or inadequate implementation [18]. The proposed alternative may not be successful, which will lead to a new analysis of the problem, evaluation of alternatives and selection of a new alternative. Thus, evaluation is a key factor in the process because decision-making is a continuous and never-ending process.

Some limitations should be stressed. First, we were unable to perform a meta-analysis due to the variation in the type of studies analysed, as well as to use a quality assessment methodology to assess the quality of studies. The sample number of eligible studies was also limited. Another limitation was regarding the characterization of the CDSS, due to some studies not directly explaining the respective approach used. There may also be some inconsistency in the review carried out due to the personal opinions of each reviewer.

It is known that in the clinical area regulations and ethical issues (e.g. computer based control of infusion pumps) limit the application of the most advanced phases of the decision-making process (e.g. choice and implementation). This means that the highest value for the maturity stage might be lower than 4. Thus, some difficulties encountered in the adoption of CDSS may be related to the medical context, such as: the development of methods for supporting choice phase of decision making; interoperability among medical devices and health care information systems; technology acceptance; ethical and regulatory restrictions; poor involvement of professionals and organizations. A lot of work should be done in the future to mitigate those limitations, starting with: i) consider a larger sample of studies; ii) determine the accepted range of values for the maturity stage in particular clinical context of application; and iii) depict what it should be the focus of research in order to fulfil the actual maturity gap and develop effective CDSS.

## Conclusion

Clinical Decision Support Systems (CDSS) represent the decision support activity and can be translated into a machine-readable computerized format. Most healthcare information systems are encouraged to or already include clinical decision support practices to organize

clinical knowledge and improve the decision-making process [3]. In order to answer the research questions formulated in this work, a systematic review was developed to identify the techniques and approaches used in CDSS from 52 studies. The outcomes were varied and did not show a main pattern, leading to limited evidence. Nevertheless, it was identified the top three trends of the four defined groups: knowledge management, technological features, type of system, and system integration. In addition, this study allowed a more complete analysis to understand the state of maturity of the CDSS. A crossover of the identified trends was made to identify the maturity regarding the four phases proposed by Simon. The results demonstrated the lack of maturity of the CDSS presented by the reviewed studies.

Decision making is a process of making a choice between several alternatives to achieve a desired outcome. Among these possible causes, the most common and serious error is an inadequate definition of the problem. When the problem is not defined correctly, the alternative selected and implemented will not produce the desired result. That said, we believe that the biggest difficulties on the CDSS adoption are in the operational clinical environment, with the involvement of characteristics, processes, and stakeholders. Furthermore, based on the identified gap in this study, an agenda should be created for what an effective CDSS should be. There is an incentive for the scientific community to contribute to mitigate the limitations in order to achieve more effective Clinical Decision Support Systems.

## Supporting information

**S1 File. Information extracted from the review.**
(DOCX)

**S2 File. PRISMA checklist.**
(DOC)

**S3 File. PRISMA flow diagram of study selection.**
(DOC)

## Author Contributions

**Conceptualization:** Francini Hak, Tiago Guimarães, Manuel Santos.

**Data curation:** Francini Hak, Tiago Guimarães.

**Formal analysis:** Francini Hak, Tiago Guimarães.

**Investigation:** Francini Hak, Tiago Guimarães.

**Methodology:** Francini Hak, Tiago Guimarães.

**Project administration:** Manuel Santos.

**Supervision:** Manuel Santos.

**Validation:** Manuel Santos.

**Writing – original draft:** Francini Hak.

**Writing – review & editing:** Francini Hak, Tiago Guimarães, Manuel Santos.

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
