## [Decision Letter · Decision Letter 0]

22 Mar 2022

PONE-D-20-38355Towards Effective Clinical Decision Support Systems: A Systematic ReviewPLOS ONE

Dear Dr. Hak,

Thank you for submitting your manuscript to PLOS ONE. After careful consideration, we feel that it has merit but does not fully meet PLOS ONE’s publication criteria as it currently stands. Therefore, we invite you to submit a revised version of the manuscript that addresses the points raised during the review process.

We look forward to receiving your revised manuscript.

Kind regards,

Gabriele Oliva, Ph.D

Academic Editor

PLOS ONE

Journal Requirements:

4. Please upload a new copy of Figures 1 and 2 as the detail is not clear. Please follow the link for more information: https://blogs.plos.org/plos/2019/06/looking-good-tips-for-creating-your-plos-figures-graphics/" https://blogs.plos.org/plos/2019/06/looking-good-tips-for-creating-your-plos-figures-graphics/

5. We note you have included a table to which you do not refer in the text of your manuscript. Please ensure that you refer to Tables 1, 2, 3, 4 and 5 in your text; if accepted, production will need this reference to link the reader to the Table.

Additional Editor Comments (if provided):

Two reviews were obtained, both suggesting minor edits to the paper. I agree with the reviewers' evaluation and I am recommending a minor revision of the paper.

Reviewers' comments:

Reviewer's Responses to Questions

**Comments to the Author**

1. Is the manuscript technically sound, and do the data support the conclusions?

Reviewer #1: Partly

Reviewer #2: Partly

2. Has the statistical analysis been performed appropriately and rigorously? 

Reviewer #1: I Don't Know

Reviewer #2: I Don't Know

3. Have the authors made all data underlying the findings in their manuscript fully available?

Reviewer #1: Yes

Reviewer #2: Yes

4. Is the manuscript presented in an intelligible fashion and written in standard English?

Reviewer #1: Yes

Reviewer #2: Yes

5. Review Comments to the Author

Reviewer #1: The topic is quite relevant and important. Overall, the paper achieved its overall aim i.e. " to systematically review the extant empirical evidence on this topic, establish the status quo of the research, and propose an agenda for further studies" However, there are a number of issues with the methods and the reporting as documented below.

This paper provides an extensive literature review related to the key features that influences the development of effective CDSS.

Method:

• The authors argue that focused only on papers published in five online data sources. There are other resources such as valuable conferences that might be even more relevant to the study's purpose than those in grey literature.

• The search strategy is not comprehensive enough for a systematic review. The search strategy is apparently missing.

• In the information resources, it is stated that five online data sources that have a major impact on health information systems research were selected. On what basis are these five main sources (references)? Which reference (s)?

Discussion:

• The authors argue that none of the reviews have provided a global trend of a CDSS through a systematic review, but the reader might find this claim ambitious (Given that you have not searched the database anymore).

• In the discussion, in systematic review articles, the results of the article should be compared with other articles related to this paper.

The structuring of the discussion and conclusion presentation is not well-argued and not clear to me.

Please rewrite the conclusion. Start with the main findings which should contain an overview of the literature you studied.

Reviewer #2: The article carries out a systematic review of clinical decision support systems. In particular, they want to identify the articles dealing with this issue by highlighting the characteristics that distinguish them, also providing information on the purpose and the recipient.

Five open source online resources are considered. In addition, defined a methodology based on specific criteria: freely accessible articles, written in English, must provide medical support, media must be computer or technological. Two reviewers will analyze the title, abstract and keyword of the articles according to the parameters and will extract the most significant ones. The extracted elements will be read entirely and analyzed. In particular, four main features will be analyzed: the management and technical representations, technological characteristics of the system, the type of system and the integration of the system. The data were classified without providing an analysis of the analyzed characteristics and differences between the various solutions proposed. It also shows the trend of the distribution and the use of different methodologies not analyzing the data or providing a comment of the same.

Authors should express more clearly the choice of criteria and the analysis made. In addition, the differences and aspects highlighted by the collection should be highlighted and clarify.

6. PLOS authors have the option to publish the peer review history of their article (what does this mean?). If published, this will include your full peer review and any attached files.

Reviewer #1: **Yes: **Maryam Zahmatkeshan

Reviewer #2: No

---

## [Author Response · Author response to Decision Letter 0]

20 Jul 2022

Reviewer #1: 

First of all, we would like to thank for the comments and constructive suggestions that will certainly contribute to the enrichment of this study. In this revised version, we reinforce all the points suggested by the reviewer, especially in the methods and in the discussion part. The changes are highlighted in yellow in the revised manuscript with track changes version.

In the 'Information Sources' chapter, we reinforce the criteria that were applied to choose data sources, based on the Scimago ranking. The search strategy was also completed, reinforcing the filters and terms used for the search. We also combine the search strategy with the eligibility criteria. In the discussion part, we refer to other systematic review studies that address the Clinical Decision Support System (DSS), but that do not relate to the objective of this study. They tend to analyse the application/performance of DSS into specific clinical problems, while we intend to analyse and characterize DSS by their capacities supporting the phases decision making process.

In the Discussion and Conclusion chapters, we rewrote as suggested, presenting a clearer, literature-based structure. We hope that the changes made are as expected.

Reviewer #2:

First of all, we would like to thank for the comments and constructive suggestions that will certainly contribute to the enrichment of this study. In this revised version, we reinforce all the points suggested by the reviewer, especially in the methods part. The changes are highlighted in green in the revised manuscript with track changes version. In the Data Extraction and Management chapter, we describe all the information that was extracted from the reviewed articles. In addition, we highlighted what was expected to be found at each stage of the decision-making process. We included the list of conditions considered for classifying DSS according to these phases (Table 2). We hope that the changes made are as expected.

---

## [Decision Letter · Decision Letter 1]

28 Jul 2022

Towards Effective Clinical Decision Support Systems: A Systematic Review

PONE-D-20-38355R1

Dear Dr. Hak,

We’re pleased to inform you that your manuscript has been judged scientifically suitable for publication and will be formally accepted for publication once it meets all outstanding technical requirements.

Kind regards,

Gabriele Oliva, Ph.D

Academic Editor

PLOS ONE

Additional Editor Comments (optional):

Both reviewers recommend acceptance. I agree with their judgement and I recommend acceptance as well.

Reviewers' comments:

Reviewer's Responses to Questions

**Comments to the Author**

1. If the authors have adequately addressed your comments raised in a previous round of review and you feel that this manuscript is now acceptable for publication, you may indicate that here to bypass the “Comments to the Author” section, enter your conflict of interest statement in the “Confidential to Editor” section, and submit your "Accept" recommendation.

Reviewer #1: All comments have been addressed

Reviewer #2: All comments have been addressed

2. Is the manuscript technically sound, and do the data support the conclusions?

Reviewer #1: Yes

Reviewer #2: Yes

3. Has the statistical analysis been performed appropriately and rigorously? 

Reviewer #1: Yes

Reviewer #2: N/A

4. Have the authors made all data underlying the findings in their manuscript fully available?

Reviewer #1: Yes

Reviewer #2: Yes

5. Is the manuscript presented in an intelligible fashion and written in standard English?

Reviewer #1: Yes

Reviewer #2: Yes

6. Review Comments to the Author

Reviewer #1: Authors have adequately addressed my comments raised in a previous round of review and my feel that this manuscript is now acceptable for publication.

With respect

Reviewer #2: The article carries out a systematic review of clinical decision support systems. In particular, they want to identify the articles dealing with this issue by highlighting the characteristics that distinguish them, also providing information on the purpose and the recipient.

Five open source online resources are considered. In addition, defined a methodology based on specific criteria: freely accessible articles, written in English, must provide medical support, media must be computer or technological. Two reviewers will analyze the title, abstract and keyword of the articles according to the parameters and will extract the most significant ones. The extracted elements will be read entirely and analyzed. In particular, four main features will be analyzed: the management and technical representations, technological characteristics of the system, the type of system and the integration of the system. Authors express clearly the choice of criteria and the analysis made. In addition, the differences and aspects highlighted by the collection are express.

7. PLOS authors have the option to publish the peer review history of their article (what does this mean?). If published, this will include your full peer review and any attached files.

Reviewer #1: No

Reviewer #2: No

---

## [Editor Report · Acceptance letter]

5 Aug 2022

PONE-D-20-38355R1 

Towards Effective Clinical Decision Support Systems: A Systematic Review 

Dear Dr. Hak:

I'm pleased to inform you that your manuscript has been deemed suitable for publication in PLOS ONE. Congratulations! Your manuscript is now with our production department. 

Kind regards, 

on behalf of

Dr. Gabriele Oliva 

Academic Editor

PLOS ONE